# Temperature Effects on Optical Trapping Stability

**DOI:** 10.3390/mi12080954

**Published:** 2021-08-12

**Authors:** Dasheng Lu, Francisco Gámez, Patricia Haro-González

**Affiliations:** 1Nanomaterials for Bioimaging Group, Departamento de Física de Materiales, Universidad Autónoma de Madrid, Calle Francisco Tomás y Valiente, 7, 28048 Madrid, Spain; dasheng.lu@estudiante.uam.es; 2Departamento de Química Física, Universidad de Granada, Avenida de la Fuente Nueva, s/n, 18071 Granada, Spain; fgammar@gmail.com

**Keywords:** temperature, optical trapping, optical forces, Brownian motions

## Abstract

In recent years, optically trapped luminescent particles have emerged as a reliable probe for contactless thermal sensing because of the dependence of their luminescence on environmental conditions. Although the temperature effect in the optical trapping stability has not always been the object of study, the optical trapping of micro/nanoparticles above room temperature is hindered by disturbances caused by temperature increments of even a few degrees in the Brownian motion that may lead to the release of the particle from the trap. In this report, we summarize recent experimental results on thermal sensing experiments in which micro/nanoparticles are used as probes with the aim of providing the contemporary state of the art about temperature effects in the stability of potential trapping processes.

## 1. Introduction

Optical tweezers (OTs) have been revealed as a versatile tool that enable the manipulation of micro- and nano-objects at the single particle level [1,2,3,4,5]. This non-contact technique covers a wide range of fields, from material studies to single-cell manipulation [6,7,8]. In a typical optical trapping experiment, a near-infrared laser beam is focused close to the diffraction limit using a high numerical aperture lens. The laser beam exerts a net force (combination of radiation pressure and gradient forces) on objects with a refractive index larger than that of the surrounding medium. In such situations, there is an effective potential minimum near the laser focus that serves as a three-dimensional trap for the particle. The stability of an optically trapped object is size-limited, and in the case of nanoparticles, its reduced size leads to optical potential forces comparable to thermal energy [9,10]. On paper, this fact limits their use above room temperature as a thermal probe, although there are several examples in the literature which avoid this limitation [11,12,13,14]. 

For instance, with the development of nanophotonics, micro/nanoparticles have emerged as a reliable probe for contactless thermal sensing [15,16,17,18]. Among all of them, luminescent micro/nanothermometers have been highlighted due to the dependence of their luminescence on environmental conditions. Although in most of the sensing experiments the probes are incorporated without a true control on their localization, the advances in optical manipulation techniques enable a single-particle incorporation into the system under study and use it as a temperature-sensing entity along the volume under investigation [3,19,20]. The aim of this review is to provide the contemporary state of the art of the temperature effects in trapping processes, showing some relevant examples on thermal sensing experiments by using single and dielectric micro/nanoparticles.

## 2. Forces on Optical Traps 

As commented above, the existence of radiation pressure and gradient force enable the trapping and manipulation of single objects at both the micro- and nanoscale. The mathematical treatment of optical forces enables us to evaluate the experimental conditions for an optical trapping setup for a target particle to be stable, i.e., when the radiation-derived forces are balanced for a given object size. The particle size parameter, ζ, establishes the limits in which either the “classical” ray optics or polarizability-based regimes are obeyed; therefore, theoretical force calculations facilitate the design of specific trapping experiments in which the light properties are selected appropriately for the stable trapping of a given target object. The value of ζ is proportional to the D/λ ratio, with D being the particle effective diameter and λ being the light wavelength. When handling micro-objects (Mie regime), ζ >>1 and the geometrical optics approach provides an acceptable route for the evaluation of optical forces, as schematically shown in Figure 1a. For spherical particles of refractive index *n*_p_ in the ray optics regime, the complete expressions for the scattering and gradient forces that suffer the particle in a homogeneous and non-dispersive medium with refractive index *n_m_* < *n*_p_ within a circularly polarized light are given by [21]:(1)$$F_{scat}=\overset{\infty }{\underset{i=1}{\sum }}\left(\frac{n_{m}P_{i}}{c}\right)\left\{1+R_{i}\cos\left(2{\hat{\theta }}_{i}\right)-\frac{T_{i}^{2}\left[\cos\left(2{\hat{\theta }}_{i}-2{\hat{r}}_{i}\right)+R_{i}\cos\left(2{\hat{\theta }}_{i}\right)\right]}{1+R_{i}^{2}+2R_{i}\cos\left(2{\hat{r}}_{i}\right)}\right\}=\overset{\infty }{\underset{i=1}{\sum }}\left(\frac{n_{m}P_{i}}{c}\right)Q_{scat,i}$$
(2)$$F_{grad}=\overset{\infty }{\underset{i=1}{\sum }}\left(\frac{n_{m}P_{i}}{c}\right)\left\{R_{i}\sin\left(2{\hat{\theta }}_{i}\right)-\frac{T_{i}^{2}\left[\sin\left(2{\hat{\theta }}_{i}-2{\hat{r}}_{i}\right)+R_{i}\cos\left(2{\hat{\theta }}_{i}\right)\right]}{1+R_{i}^{2}+2R_{i}\cos\left(2{\hat{r}}_{i}\right)}\right\}=\overset{\infty }{\underset{i=1}{\sum }}\left(\frac{n_{m}P_{i}}{c}\right)Q_{grad,i}$$
where $P_{i}$ is the power of the ith ray and $T_{i}$ and $R_{i}$ are the polarization-dependent Fresnel coefficients for transmission and reflection, respectively. ${\hat{\theta }}_{i}$ and ${\hat{r}}_{i}$ are the incidence and transmission angles and *c* is the speed of light in a vacuum. The dimensionless trapping efficiencies $Q_{grad,i}$ and $Q_{scat,i}$ account for the momentum transferring efficiency and determine the trapping efficiency of a certain ray as:(3)$$Q_{trap}=\sqrt{Q_{scat}^{2}+Q_{grad}^{2}}$$

When handling nano-objects (Rayleigh regime), ζ << 1, the distinction between the components of reflection, refraction and diffraction can be ignored. The perturbation of the incident wavefront is minimal; therefore, the particle can be viewed as an induced dipole that behaves according to simple electromagnetic laws (see schematic diagram in Figure 1b), and the scattering force is due to the absorption and reradiation of light by the dipole. The gradient force is that experienced by a dipole in an inhomogeneous electric field (i.e., the focused laser beam). This force points in the direction of the gradient electromagnetic field, attracting the object towards the optical trap center. The scattering force points in the direction of propagation of the incident light and is proportional to the light intensity. For a sphere of radius *a* trapped under a light with intensity *I*_0_, these forces are:(4)$$F_{scatt}=\frac{128\pi^{5}a^{6}}{3\lambda^{4}}\left(\frac{I_{0}n}{c}\right){\left(\frac{m^{2}-1}{m^{2}+2}\right)}^{2}$$
(5)$$F_{grad}=\left(\frac{2\pi a^{3}}{c}\right)\left(\frac{m^{2}-1}{m^{2}+2}\right)\nabla I_{0}$$
where *m* refers to relative refractive index of trapped particles to the medium, *m* = *n_p_*/*n_m_*.

When the particle size is comparable to the trapping light wavelength, ζ∼1, a full electromagnetic solution of the problem is required. The radiation forces are then evaluated from the conservation of linear and angular momentum in the light–matter interaction though the time-averaged Maxwell tensor, given for monochromatic radiation by:(6)T¯=12εmRe[Et⨂Et*+c2n2Bt⨂Bt*−12(|Et|2+c2n2|Bt|2)I]
where $E_{t}\;$ and $B_{t}$ are the total (incident plus scattered) electric and magnetic fields, respectively, and $\varepsilon_{m}$ is the dielectric constant. ***I*** is the unit dyadic and ⨂ is the dyadic product. Most objects that are useful or interesting in optical trapping, in practice, tend to fall into this intermediate size range (0.1–10 λ), but as a practical matter, it can be difficult to work with objects smaller than video microscope resolutions (~0.1 μm), although particles as small as ~16 nm in diameter have successfully been trapped [22].

## 3. Force Calibration and Local Conditions Effects

### 3.1. Force Calibration Methods

Regardless of the particle size, the experimental calibration (i.e., transduction signal-to-real unit conversion) of the optical forces is usually performed with different methodologies that are traditionally divided into passive and dynamic methods. In passive methods, the particle trajectory is recorded in a fixed trap, whereas dynamic methods rely on the induction of a drag force on the trapped particle by a flowing medium $\frac{\kappa }{2\pi k_{B}}$. For instance, for the measured trajectory of a trap object, its analysis can be performed under different frameworks.

If one assumes that a position distribution obeys a Maxwell–Boltzmann functionality under the harmonic trapping potential and that the equipartition-theorem is valid, the potential analysis [23,24] and the equipartition methods are applicable, and they are described as follows. In the first case, by fitting the statistical distribution of microparticle positions p(X) to a Maxwell–Boltzmann probability distribution:(7)$$p\left(X\right)=p_{0}\exp\left(-U/k_{B}T\right)$$
where *k_B_* is the Boltzmann constant and *T* is the absolute temperature. From this equation, the trap potential U can be evaluated without assuming harmonic behavior. In the equipartition-theorem analysis, one assumes the potential is harmonic and with a trap stiffness κ; hence:(8)$$\langle U\rangle =\frac{1}{2}\kappa \langle {\left(X-X_{0}\right)}^{2}\rangle$$
where $X_{0}$ is the equilibrium position and the brackets stand for the ensemble averages. According to the Boltzmann statistics, the probability of finding the particle in a certain position when subjected to a harmonic potential well follows a Gaussian distribution:(9)$$p\left(X\right)={\left(\frac{\kappa }{2\pi k_{B}}\right)}^{1/2}\exp\left[-\frac{\kappa {\langle (X-X_{0})\rangle }^{2}}{2k_{B}T}\right]$$

The thermal fluctuations of a trapped object can then also be used to obtain the trap stiffness through the equipartition theorem.
(10)$$\frac{1}{2}k_{B}T=\frac{1}{2}\kappa {\langle (X-X_{0})\rangle }^{2}$$

Thus, by measuring the mean squared displacement (MSD) of a trapped object, the stiffness can be determined. To do this, all positions visited by the trapped particle for a certain period of time must be measured with enough temporal and spatial resolution. Alternatively, the decay time of the particle autocorrelation function [25,26] or the mean square displacement analysis also provide trap stiffness and friction coefficients values. More recent approaches are represented by maximum likelihood estimation analysis [27].

Certainly, the most frequent passive analysis is the power spectra method [28] that rest on the Einstein–Ornstein–Uhlenbeck description of Brownian motion [29], which describes the dynamics of a particle of mass *m* and temperature *T* in a harmonic potential by the motion equation:(11)$$mX^{\prime \prime }\left(t\right)+\kappa X+\gamma X^{\prime }\left(t\right)={\left(2kT\gamma \right)}^{\frac{1}{2}}\xi \left(t\right)$$
where γ is the drag coefficient and ξ(*t*) is a stochastic or Langevin force of zero mean. In most cases, the inertial term can be neglected, and the time-to-frequency Fourier-transform of Equation (11) provides the power spectrum density (PSD) of the particle position in the trap.
(12)$$<\Delta x^{2}\left(f\right)>=\left(\frac{2\gamma kT}{\kappa^{2}}\right)\cdot \frac{f_{c}^{2}}{f_{c}^{2}+f^{2}}$$
where $f_{c}=\kappa /\left(2\pi \gamma \right)$ is the corner frequency, above which the motion is purely diffusive. The Lorentzian fit of the PSD provides an exact value for the corner frequency, and the PSD-based force can easily be given by:(13)$${<\Delta F^{2}\left(f\right)>}_{PSD}=\kappa^{2}<\Delta x^{2}\left(f\right)>$$

At this point, it should be mentioned that the solution of the complete set of Langevin equations for a spherical particle in harmonic potentials enables distinguishing between different timescales that determine the particle behavior (see in Figure 2):$\;\tau_{p}=m^{*}/\gamma$, $\tau_{f}=\rho_{f}a^{2}/\eta$, $\tau_{k}=\gamma /\kappa$. $\rho_{f}$ and η are the density and viscosity of the fluid, respectively, and $m^{*}=m+m_{f}$ is a modified mass influenced by hydrodynamics, where $m_{f}$ is the mass of the displaced fluid [30,31]. The first two values are related to fluid vortex propagation and inertial timescales, whereas $\tau_{k}$ measures the ratio between the Stokes friction coefficient and the optical trap stiffness. For timescales much shorter than $\tau_{k}$, the Brownian motion is considered to be free, i.e., it is the perpetual and random movement of particles suspended in a fluid. Normally, $\tau_{p}$ is smaller than $\tau_{f}$, although $\tau_{f}$ could be larger or smaller than $\tau_{k}$ depending on the magnitude of the trap stiffness. At a very short timescale ($t\ll \tau_{p}$), the Brownian motion of the particle is in the so-called ballistic regime, and is thus dominated by the particle mass; therefore, $MSD=(k_{B}T∕m^{*})t^{2}$. The particle movement is subjected to the hydrodynamic memory effect of the liquid at an intermediate timescale ($\tau_{p}<t<\tau_{f}$). Finally, at a longer time scale ($t>\tau_{f}$) the Brownian motion is generally thought to be governed by the particle diffusion coefficient $D_{f}$, with $MSD=2D_{f}t$ according to the Einstein relation. At a longer timescale ($t>\tau_{k}$), the optical trap is dominant, and the MSD shows a plateau with $MSD=2k_{B}T/\kappa$ [32], which would overlap with the diffusion region ($t>\tau_{f}$) in some cases. Thanks to recent improvements in position detector and time-resolved technologies, the direct observation of the ballistic-to-diffusive Brownian motion transition for optically trapped particles in a liquid has been achieved [30,33,34].

### 3.2. Effects of Local Conditions on Trapping Forces

However, most of these methods rely on the comfortable assumption that the medium viscosity and/or trap local temperature are known. In viscoelastic media, at least, the former situation is definitely not fulfilled, and the mechanical response of the (dense) media has to be incorporated into the Langevin equation. Consequently, the calibration from the evaluation of thermal fluctuations requires additional inputs. To overcome this difficulty, Fischer and Berg-Sørensen [35] developed a calibration procedure based on the fluctuation theorem, called the microrheology method, that combines the measurement of the thermal motion of the trapped particle with its positional response against an oscillating stage motion at different frequencies. The combination of this theorem with the analysis of both thermal and mechanical response effects on the PSD enables extraction of the calibration constant, the trap stiffness, and the viscoelastic properties of the surrounding media. Other alternatives are also available, as the robust measurement of light momentum [36] that bypasses the nonlinear effects in the harmonic potential model in which the other calibration alternatives are sustained.

Although the trap stiffness, and thus the calibration of force measurement, do not depend on the temperature but on many optomechanical details of the setup and the size and shape of the trapped bead, the calibration constant extracted from most of these methods assume a known friction factor, i.e., the particle size and viscosity are both fixed. This is not the case in a heating trap, and differences in the calibration constant for different acquisition methods have been observed [37]. On the one hand, the effect of the temperature on the solvent has a direct influence on the microscale motion and the optical trapping efficiencies; an established generalized protocol for determining temperature increments in optical traps has not been reported to date. To that aim, luminescence thermometry using molecular [38] and nanoprobes [6,39], non-absorbing particles [40], Stokes shifts of fluorescence [41] or refractive index measurements [42] have been reported. Some experimental routes aiming at both controlling the trap temperature or minimizing the heating in the focal coordinate have been described. Among the first set of methods, one can find the decoupling of trapping and heating lasers [43,44] or direct external controls [45]. Shifting the wavelength of the trapping radiation or modulating the laser power in femtosecond tweezers [43] or in counterpropagating setups [46] are also used. Interestingly, laser-induced heating is the core of some research lines in which, for instance the convection currents are employed to manipulate microparticles in the low-power regime [47] or to enhance the trapping efficiency [48]. On the other hand, this effect is outgrown when the trapped particles have an appreciable absorption cross-section at the trapping wavelength and trap-heat decoupling is not possible because of a broad absorption spectrum of the trapped object. Force calculations employing the aforementioned equations are strictly valid for non-absorbing particles; therefore, forces might be modified upon local photothermal conversion upon photon absorption or by direct heating of the surrounding media by an external source or by the laser-induced heating of the solvent itself just commented. In such cases, convective fluxes become ever more non-negligible, and the acute solvent viscosity gradient are imposing information for a proper description of particle trapping. Overall, the problem might be understood in the theoretical framework of a “hot Brownian motion”. In ref. [49], Rings et al. presented a method to account for the temperature effect on the Brownian motion of an arbitrary yet spherical particle by the introduction of an effective viscosity and particle size. Such an approach was further developed for anisotropic particles [50] in which the laser polarization may induce alignments or rotations.

## 4. Temperature Effect on Optical Trapping

### 4.1. Temperature Effect on the Optical Trapping of Microparticles

According to the description of Brownian motion in an optical trap in Figure 2, at a longer timescale ($t\gg \tau_{k}$), trapped particles are supposed to be confined within an area of radius $R_{p}\sim \sqrt{2k_{B}T/\kappa }$. When $R_{p}$ is greater than the trap radius $R_{t}$, the trapping stability is weak, whereas if $R_{p}\leq R_{t}$, the trapping is efficient and particle manipulation can be achieved. After transformation, $\frac{1}{2}\kappa R_{t}^{2}\geq k_{B}T$ is obtained, i.e., the trapping potential or work needed to release particles from the trap ($\frac{1}{2}\kappa R_{t}^{2}$) should be larger than the thermal energy $k_{B}T$, which can be used as a criterion for the stability of optical trapping.

Microparticles can be confined within a small area thanks to the relatively high values of trapping stiffness. Some typical κ values for selected microparticles can be found in Table 1. For example, for a microparticle with a diameter of 2 μm, the experimentally measured trap stiffness is around 10 pN/μm. Considering a temperature rise from 293 K to 353 K, $R_{p}$ changes only from 28 to 31 nm (see in Table 2), which is much smaller than the radius of conventional optical traps (i.e., the optical trap generated by a focused 980 nm laser beam using an objective 100× with 0.85 NA, $R_{t}=0.61\lambda /NA=700$ nm). In these cases, the influence of thermal energy can be ignored; thus, the stable trapping and manipulation of microparticle can be achieved even at high temperatures. As an exemplification of the temperature effect on the Brownian motion of microparticles in a trap, Brownian dynamics simulations at different temperatures were performed using Brownian Disks Lab (BDL) software [32,51]. Brownian Disks Lab (BDL) is an open-source software developed in Easy Java/Javascript Simulations(EjsS) [51] for the evaluation of trajectories of two-dimensional Brownian disks under external forces by means of Brownian dynamics simulations without hydrodynamic interactions [2] employing the Langevin formalism Equation (11) in the overdamped formulation: $X\left(t\right)=\frac{1}{\gamma }(-\kappa X+{\left(2k_{B}T\gamma \right)}^{\frac{1}{2}}\xi \left(t\right))$. Setting the diameter to 2 μm and the trap stiffness to 10 pN/μm, with different bath temperatures (293 and 353 K) as input parameters, the probability density distributions from Brownian simulations are shown in Figure 3a. As can clearly be observed, the temperature has little effect on the distribution function of the trapped microparticle and on the calculated trap stiffness (see in Table 2). In other words, even when the instantaneous velocity increases when the temperature rises [52], the microparticle still moves in an area that is only slightly expanded, and the trapping stability of microparticles is little affected by temperature. From statistical grounds, the variance of the distribution enables evaluation of the trap stiffness κ by using: $\sigma^{2}=\frac{k_{B}T}{\kappa }$. The standard deviation of the distribution is proportional to the square root of the temperature; therefore, temperature has little effect on the distribution of trapped particles. In practice, light absorption by the surrounding medium or by the trapped particle itself can lead to increments of the local temperatures in the trapping volume of up to 40 °C [53,54]. This is the reason why the manipulation and assembly of microparticles in fluids are widely used in optical force calibration and 3D scanning thermal imaging by an optically trapped temperature probe [55,56].

### 4.2. Temperature Effect on Optical Trapping of Nanoparticle

Traditional OT applied to the manipulation of nanosized objects, such as Rayleigh nanoparticles and biomolecules, found some limitations, such as the fundamental limit imposed by laser beam diffraction. The huge reduction in the trapping strength becomes insufficient for the optical stabilization of nanoparticles that lead to an erratic Brownian motion. Hence, nanoparticles cannot be confined within the trap volume with traditional techniques, and it becomes a challenge to manipulate and trap nanoparticles stably even at room temperature. For an 8 nm nanoparticle, the experimentally determined trap stiffness is around 0.01 pN/μm (see Table 1). At room temperature $R_{p}\sim 900$ nm, which surpasses the radius of conventional optical traps (700 nm), indicating that at a longer timescale ($t\gg \tau_{k}$), the nanoparticle will escape from the trap. To explore the position distribution of a nanoparticle in the trap volume, the same Brownian dynamics simulations were performed by setting the trapped nanoparticle diameter to 10 nm and the trap stiffness to 0.01 pN/μm. The probability density distributions of the trapped nano-object in the X coordinate at two temperatures are presented in Figure 3b, and the comparison of the two-dimensional (2D) trajectories of a single microparticle and nanoparticle in a trap can be found in Figure 3c. Obviously, nanoparticles are allowed to diffuse in regions far beyond the trap volume (a circle with a radius of 700 nm). A simple statistical analysis of the probability density distributions of the X coordinate enables evaluation of the probability for a 10 nm nanoparticle to escape from the simulated trap (see Table 2). At room temperature, this escaping probability was calculated to be 27%, which is the main factor leading to the release of nanoparticles from an optical trap. This probability intuitively increases with temperature, and consequently, the thermal stability of optical trapping further decreases at higher temperature. In addition, according to the Stokes–Einstein prediction [52], as the particle size decreases, the Brownian velocity increases dramatically. Compared with trapping microparticles, the instantaneous velocity of Brownian motion becomes a non-negligible factor during the release of trapped nanoparticles, especially when the nanoparticles are close to the optical trap boundary. This effect has been demonstrated in some of our previous experiments in which the drag method was used to calibrate the optical force exerted on trapped 257 nm nanoparticles at controlled temperatures ranging from 20 to 70 °C (see in Figure 4a). The experimental trap stiffness as a function of the temperature presented a downward trend (see in Figure 4b) because the drag force deviated the trapped nanoparticles from the trap center to positions close to trap boundaries. Sometimes, even if the drag force is not enough to release the particles, the nanoparticles will still escape from the trap with the assistance of the Brownian motion velocity. Experiments have indeed demonstrated that the trapping stability of nanoparticles is sensitive to temperature, as shown in Figure 4c,d: an 8 nm nanoparticle escapes from the trap at temperatures around 30 °C, which proves again that the temperature dependence of the Brownian motion velocity plays an important role in the release of trapped nanoparticles. Therefore, for nanoparticles, the temperature has a determinant effect on the trapping stability, and it becomes urgent to improve the trapping forces and expand the application of optical trapping to the nanoscale. Considering the disturbance induced by instantaneous velocity effects, the trapping potential should be much larger than thermal energy, as presented in ref. [59], the ratio of the trap energy to the thermal energy should be 10 or greater for the nanoparticles to remain in the trap.

In this sense, there are several effective approaches to enhance the optical forces involved in the manipulation of Rayleigh particles [4,11,13,14,22,60,61,62,63,64,65,66,67]. According to the expression of the gradient force in the Rayleigh regime, these methods can be divided into two categories. One is based on the reduction in the optical trap volume, by using nano-devices such as plasmonic tweezers [60,61,62,63], slot waveguides [14,65] and photonic crystal resonators [13,64]. Within these methods, incident laser beams have been confined within a small region well below the diffraction limit, exerting a sufficiently strong force to manipulate the nano-objects. However, these procedures require complex nanofabrication processes and setup customization; thus, they are not easy to implement for certain applications such as in biosensing, employing commercial equipment. Moreover, relatively bulky and expensive optical elements such as high numerical aperture (NA) objective lenses are required to optimize the optical force and limit the miniaturization and integration of these devices. The reduction in the optical trap volume can also be achieved by using a dielectric microsphere (or microcylinder) as a focusing lens [4,66,67,68,69]. A photonic nanojet (PNJ) is produced on the shadow side of the dielectric microsphere when it is illuminated by a plane wave. A PNJ usually has a sub-diffraction-limit beam waist; therefore, the optical gradient force near a PNJ is much stronger than in a beam focus originating from a high-NA objective lens.

Another alternative is based on increasing the polarizability of nanoparticles, such as a dipole interacting with an electromagnetic field [2]. Optical gradient force is proportional to the magnitude of polarizability of nanoparticles in fluid. Most of the nanoparticles used for optical trapping, such as upconverting nanoparticle, are dielectric particles with low polarizability; therefore, the modification of surface charges or zeta-potential has a direct influence on the polarizability of the nanoparticles. Therefore, it is an effective method to improve optical force by modifying the nanoparticle surface [58,70] and/or the surrounding medium [71,72,73]. Additionally, extensive doping with lanthanide ions in up-converting nanoparticles is also a good route to enhance the permittivity and polarizability of nanocrystals, leading to enhanced optical trapping forces by several orders of magnitude [74].

## 5. Conclusions

Non-contact thermal sensing is now a tangible fact thanks to the joint development of luminescent micro/nanoparticles and optical manipulation techniques such as optical tweezers. A key feature that modulates the reliability of these results is not only the quality of the micro/nano-probes, but also their intrinsic stability in the optical trap. In that sense, small temperature changes in the optical trap may affect the micro/nanoparticle stability and compromise the expected results. In this review, we have analyzed the potential temperature effects in trapping processes, showing some relevant examples, and bypassing alternatives. However, this point has only been considered recently, and there is still room for improvements in both the engineering nanoprobes and in the trapping setups. Contemporary advances in this field are providing not only direct applications requiring a movable heat source (as in optical hyperthermia), but also new insights into the statistical mechanics and the definition of temperature at the single particle level.

## Figures and Tables

**Figure 1 micromachines-12-00954-f001:**
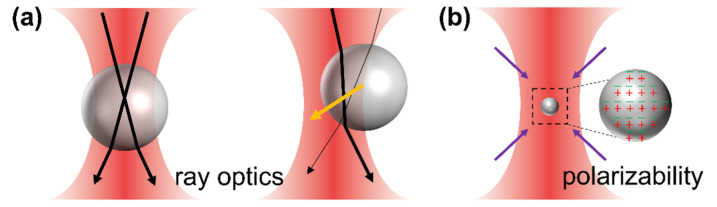
(**a**) Mie regime: schematic diagram of a trapped microparticle at the equilibrium position and off-center position. Orange arrows represent the restoring forces acting on the microparticle, and light rays are presented as black arrows (thicker rays stand for brighter rays). (**b**) Rayleigh regime: schematic diagram of a trapped nanoparticle. Purple arrows represent the forces acting on the particle.

**Figure 2 micromachines-12-00954-f002:**
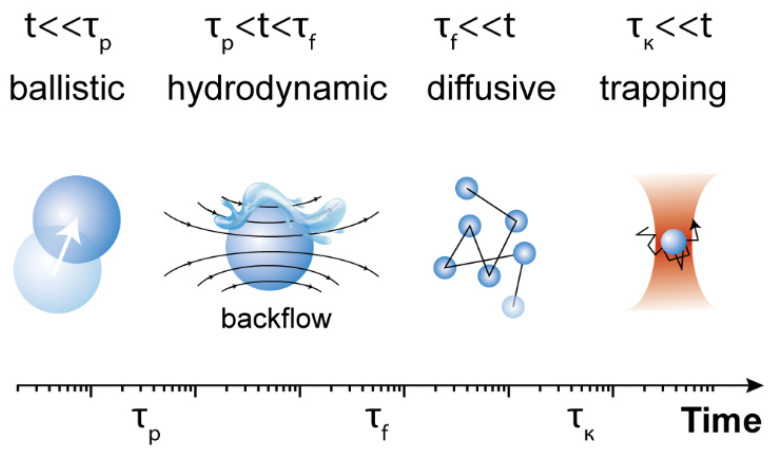
Schematic characteristic timescales of Brownian motion in different regimes.

**Figure 3 micromachines-12-00954-f003:**
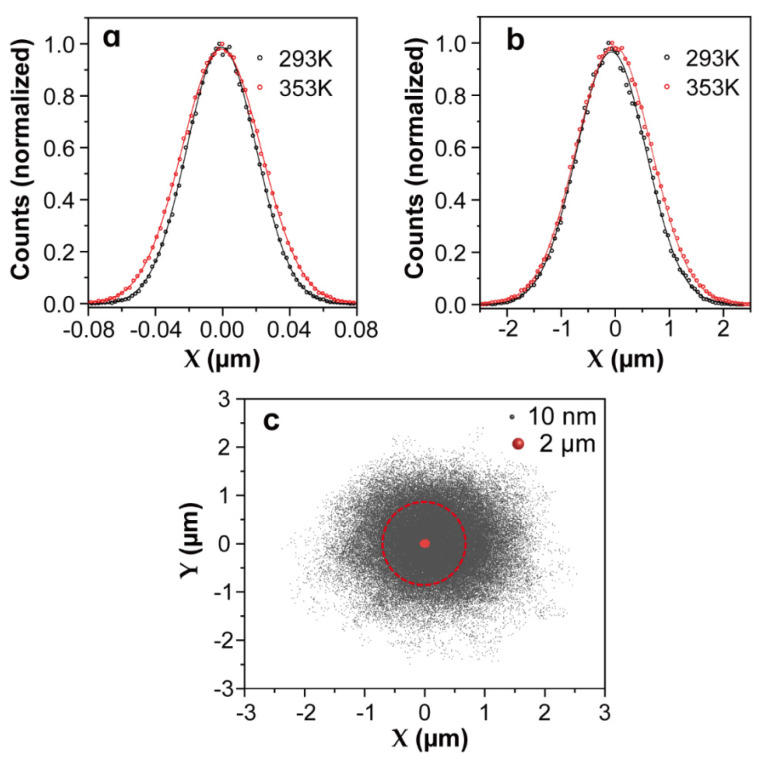
The Brownian motion dynamic simulation results of microparticles and nanoparticles at given trap stiffnesses and temperatures, which were generated by Brownian Disks Lab (BDL) softwares. Temperature dependence of probability density distributions in the X coordinate extracted from the motion of a single microparticle (**a**) or nanoparticle (**b**) under harmonic potentials. For the microparticle, the diameter and trap stiffness were set to 2 μm and 10 pN/μm, respectively, whereas for the nanoparticle, they were 10 nm and 0.01 pN/μm, respectively. The lines are non-linear fits to Gaussian functions. (**c**) Comparison of simulated two-dimensional (2D) position distributions of a single microparticle and nanoparticle optically trapped by the same focused laser beam. The red circle indicates the spot size of the optical beam with a radius of 700 nm.

**Figure 4 micromachines-12-00954-f004:**
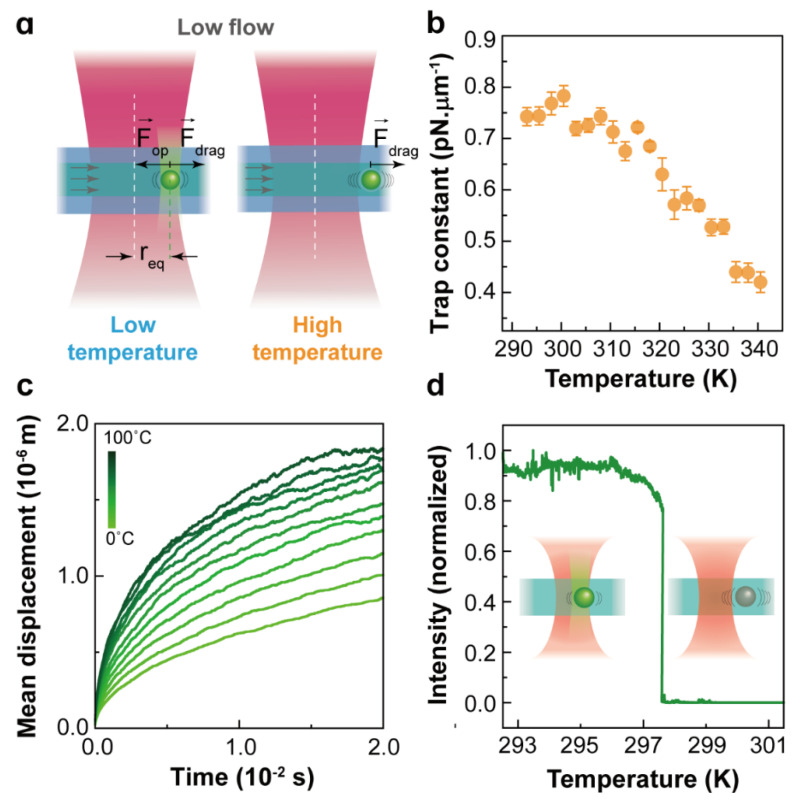
(**a**) Optical force calibration of NaYF_4_:Er^3+^,Yb^3+^ nanoparticles with a diameter of 257 nm and thickness of 143 nm at different temperatures evaluated with the drag method; (**b**) the experimentally determined trap stiffness as a function of temperature, Reprinted with permission from ref. [39]. Copyright 2020 American Chemical Society. (**c**) Simulated average value of the 8 nm nanoparticle’s mean displacement (in the X-Y plane) versus time for temperatures ranging from 0 °C to 100 °C under a 980 nm laser beam focused by an objective of (100×, 0.85 NA); (**d**) experimentally determined visible emission intensity of a trapped single 8 nm nanoparticle as a function of temperature. Here, the luminescence of nanoparticle allows the determination of whether the nanoparticle is in the trap. Modified from ref. [22]. Copyright 2021 Wiley-VCH GmbH, Weinheim.

**Table 1 micromachines-12-00954-t001:** Trap stiffness of different sizes of dielectric particles.

Material	Medium	Size (nm)	Trap Stiffness ^1^ (pN/μm)	Reference
SiO_2_	water	2000	9.5	[22]
NaYF_4_:Er^3+^,Yb^3+^	water	2000	10	[19]
NaYF_4_:Er^3+^,Yb^3+^	water	2000	2.86	[56]
NaYF_4_:Er^3+^,Yb^3+^	water	257	0.8	[39]
NaYF_4_:Er^3+^,Yb^3^	water	26	0.1	[7]
QDs (CdSe/ZnS)	water	26	0.16	[57]
SrF_2_:Ho^3+^,Yb^3+^	water	8	0.007	[22]
SrF_2_:Er^3+^,Yb^3+^	deuteroxide	8	0.033	[58]

^1^ Laser power set at 100 mW.

**Table 2 micromachines-12-00954-t002:** Statistical analysis of probability density distributions of a single 2 μm and 10 nm particle along the X coordinate at different temperatures.

Temperature/K	*R_p_*/nm		σ/nm		Calculated Trap Stiffness ^1^ κ/(pN/μm)		Probability of Being Outside (−700, 700)	
	2 μm	10 nm	2 μm	10 nm	2 μm	10 nm	2 μm	10 nm
293	28	899	20.7	643	9.3	0.0098	~0	27.6%
313	29	929	21.5	655	9.3	0.010	~0	29.3%
333	30	959	23.2	686	8.5	0.0098	~0	30.8%
353	31	987	23.6	693	8.4	0.010	~0	31.2%

^1^ Obtained by fitting the statistical position distribution along the X coordinate (in Figure 3a,b) to a Maxwell–Boltzmann probability distribution.

## Data Availability

The data presented in this study are available upon request from the corresponding author.

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
