# Peer review of "Temperature Effects on Optical Trapping Stability"

_micromachines, 2021, doi:10.3390/mi12080954_

Round 1

Reviewer 1 Report

The manuscript claims to be a review on temperature effects in optical trapping. After a good introduction the two following chapters describe basic considerations about optical forces and introduce very well know models for which exist already some reference papers. The first part of chapter four, which treats temperature effects, is mainly based on calculations using a software package which is not presented in details. The second part dealing with nanoparticles is based on two very recent papers, where on is of the authors group. Besides, quite a great number of the cited references does not treat directly temperature effects. 

In conclusion the manuscript does not meet the standards for a "Review". I apologize but I suggest to reject the manuscript for publication to Micromachines.

- The presentation of the force models is not well equilibrated. Either the description should be more detailed to discuss the underlying physics or it should be shortened. As none of the presented features are really new (excellent reference papers are existing on each approach) and as they are not very relevant for the following presentation, I would suggest the second option.

- No details about the main features of the used software are given.

- It is counter intuitive that the Gaussian position distribution is not changing with temperature. Following equation 9, the distribution width sigma is kbT/kappa. For a temperature increase from 293 K to 353 K, kbT is increasing by a factor of about 1.2. The distribution width should also increase by this factor! The values in table 2 gives however only a increase of 1.14 (1.08 for nanoparticles). This discrepancy is discussed at all!

- Why no power spectrum analysis is done? This method is described in the paper and as the high frequency response does not depend on the trap stiffness, but only on temperature effects, it would allow to separate the two features!

- The expected influence of the temperature on the trapping behavior is not clear. In the framework of the harmonic oscillator model, the trap stiffness kappa should not depend on the temperature. However, as the Brownian motion and the dynamic viscosity depends on temperature, trapped particles should move faster and over a larger range. These dependencies should be discussed in more details.

Reviewer 2 Report

Trapping with optical tweezers is gaining importance thanks to the variety of applications to the control of micro and nano particles. The review of Dasheng Lu et al. focuses on one important aspect of this technique which is usually overlooked, that is the role of temperature effects on the stability - and therefore the reliability - of optical trapping of micro/nanoparticles. After giving the theoretical background on optical forces acting on micro/nano objects and on force calibration methods, the manuscript summarizes the recent results on the topic for both micro and nano particles, and highlights possible solutions to increase the optical force acting on the particles. The review is clear, well documented and easy to follow, even for the non-expert reader thanks to the well written and comprehensively described introductory part. I find the manuscript  ready for publication after minor English spell checks and well suited to the special issue Optical Trapping of Micro/Nanoparticles.   I have a few comments for the authors to consider to improve the draft: - There is a missing and in the phrase: "account for the momentum transferring efficiency and determine the trapping efficiency of a certain ray as:" - Eq. 4-5: the paramenter m has not been defined in the text yet - Eq. 7: k is defined as Boltzman constant but later on in the text k_B is used. The choice should be uniform in all the text - The phrase "If one assumes that a position distribution obeys a Maxwell-Boltzmann functionality under the harmonic trapping potential and that the equipartition-theorem is applicable,
the potential analysis [22,23] and the equipartition methods are described, respectively" is not very clear. I suggest to end it like: "the potential analysis [22,23] and the equipartition methods are applicable and they are described in the following" - The subject is missing in: "Among the first set of methods are to decouple the trapping and heating lasers [42,43] or the direct external control [44]" - The verb is missing in: "the ratio of the trap energy to the thermal energy should be 10"  

Reviewer 3 Report

It is a nice review paper with an interesting topic. The temperature inside and around the optical trap is an inevitable factor as a highly focused laser beam is there.  However, the temperature effect is related to the real condition of the experiment so that it is difficult to be exactly measured or even estimated. This paper review the temperature effect of the optical trap, which is useful to the readers in the field of optical tweezers. For the content, I have some comments as follows.

  1. It's better to show some figures in Section 2 for understanding the force induced by optical trap.
  2. It is better to add some representative literature, like K.C. Neuman and S.M. Block, Rev. Sci. Instrum., 2004, 75: 2787. These former literature can help the readers a better understanding to this topic.
  3. It is better to add a section to comment the feature of the research point. 
